# The effects of risk-taking, exploitation, and exploration on creativity

**Tsutomu Harada** *

Graduate School of Business Administration, Kobe University, Kobe, Hyogo Prefecture, Japan

* harada@people.kobe-u.ac.jp

## Abstract

The purpose of this paper was to investigate the effects of risk-taking, exploitation, and exploration on creativity by taking a model-based computational approach to both divergent and convergent thinking as primary ingredients of creativity. We adopted a reinforcement learning framework of Q learning to provide a simple, rigorous account of behavior in the decision-making process and examined the determinants of divergent and convergent thinking. Our findings revealed that risk-taking has positive effects on divergent thinking, but not related to convergent thinking. In particular, divergent thinkers with a high learning capacity were more likely to engage in risk-taking when facing losses than when facing gains. This risk-taking behavior not only contributes to the rapid achievement of learning convergence, but is also associated with high performance in divergent thinking tasks. Conversely, both exploitation and exploration had no significant effects on creativity once these risk attitudes were considered. Moreover, while convergent thinking relied on personality characteristics, it was not associated with risk-taking, exploitation, or exploration.

**Data Availability Statement:** The data is now at osf.io/9sej6

**Funding:** This work was supported by JSPS KAKENHI under Grant (Number 26380506).

## 1. Introduction

Although creativity has long fascinated scientists, especially in the fields of neuroscience and cognitive psychology, more theoretical and empirical work is required for a thorough understanding of the mechanism and determinants of creativity [1, 2]. The purpose of this paper was to investigate the determinants of creativity by taking a model-based computational approach to creativity. We adopted a reinforcement learning (RL) framework to provide a simple, rigorous account of behavior in the decision-making process. The RL framework is supported by considerable empirical evidence, including neural signals in various cortical and subcortical structures that behave as predicted [3–6]. The framework has been applied to studies of decision making and learning in various social contexts [7–13]. However, little attention has been paid to creative aspects of decision making.

This paper assumed that a Q learning model in the RL framework underlies creative thinking and attempted to relate its properties to creativity performance. More specifically, we estimated parameters of Q learning such as risk-taking and inverse temperature based on the results of the Iowa Gambling Task (IGT). We investigated empirically how risk-taking and inverse temperature (exploitation/exploration) account for performance in creative tasks by

**Competing interests:** There are no conflicts of interest to declare.

controlling for several related cognitive and psychological factors. Thus, the Q learning model was related to creative thinking through risk-taking, exploitation and exploration in this study.

Several studies have revealed that creative people are energized and excited by challenging and risky problems [14, 15], which can lead to the experience of 'flow' [16]. These findings suggest that creativity is closely related to risk-taking. Indeed, several recent studies have investigated the relationship between risk-taking and creativity [17–23], yet the findings have been inconclusive. The differences in results could be accounted for by diversity in the research measures, definitions of risk-taking, and cultural backgrounds of the participants [24]. Furthermore, these studies commonly adopted questionnaire-based risk measurements, although the questionnaires used varied amongst the studies. Conversely, in the present study, we measured risk attitudes via the estimation of underlying utility function. We were interested in examining how the results would differ with different risk measures.

In addition to risk-taking, we also examined the effects of exploitation and exploration on creativity. Exploitation refers to the optimization of current tasks under existing information and memory conditions, while exploration implies wider and sometimes random search and trials that do not coincide with the optimal solutions provided by exploitation (see [25], for the trade-off between exploitation and exploration in the RL framework). Creativity requires both exploration and exploitation. In exploration, a wider search for a greater range of information is undertaken. In some cases, previously acquired knowledge must be unlearned so as not to get stuck with current knowledge, constraints or implicit assumptions, making it harder for them to 'think outside the box'. At the same time, creativity also relies on exploitation because the efficiency of search in a much narrower space should take full advantage of existing information. Thus, both exploration and exploitation appear to be advantageous in creative thinking, although the relative weight of each depends on the phase of the creative thinking process.

In the Q learning model, exploitation implies the selection of choices yielding the highest Q values, whereas exploration involves a preference for other non-optimal choices. These factors can be represented via an inverse temperature scale using the softmax function, as described in the Methods section. On the scale, a higher (lower) value implies a greater (lower) emphasis on exploitation (exploration). This paper exploited this measure to examine the effects on divergent and convergent thinking performance in creative tasks. In so doing, we aim to shed new light on the underlying mechanism of creative thinking.

In so doing, it is important to distinguish between divergent and convergent thinking. While measures for creativity have been widely adopted in the neurosciences as a unitary and coherent construct by equating creative thinking with divergent thinking [2, 26], convergent thinking could also account for creative outcomes because a new idea must be tested as a candidate for solution to a practical problem. For example, convergent thinking is sometimes instrumental in generating insight during problem solving [27]. Accordingly, a number of recent studies investigated the difference between the two types of thinking [23, 28]. Following these studies, this paper examined the effects of risk-taking, exploitation, and exploration on both divergent and convergent thinking.

## 2. Methods

### 2.1. Participants

A sample of 113 healthy undergraduate students at Kobe University (49 females, age range = 18–20 years, SD = 0.66) participated in the study. All participants were native Japanese with normal or correct-to-normal vision. The local Ethics Committee at the graduate school of business administration, Kobe University, approved this study, and all participants and their academic advisers signed an informed consent form before the experiment and were paid JPY

3,000 (approximately USD 28).

## 2.2. Q learning model

We adopted a variant of Q learning model [25] to account for decision making in the IGT [29]. Participants make a series of 100 choices from four decks of cards. Two of the decks are advantageous and two of them are disadvantageous. The two disadvantageous decks always give rise to relatively high gains ($100) but also to occasional large losses ($150) with a 50% chance, which result in an average loss of -$25 per trial. The two advantageous decks always generate lower gains each time ($50) but produce no losses ($0) with a 50% chance, resulting in an average gain of +$25 per trial. The goal is to maximize net scores across trials.

At each trial t, the action value $Q_i(t)$ of the chosen option (deck) i is updated via the following rule:

$$Q_i(t+1) = \begin{cases} Q_i(t) + \alpha_t^+ \delta(t) + \phi \; if \; \delta(t) \geq 0, \\ Q_i(t) + \alpha_t^- \delta(t) + \phi \; if \; \delta(t) < 0, \end{cases} \tag{1}$$

with

$$\delta(t) = U(R_i(t)) - Q_i(t), \tag{2}$$

$$U(R_i(t)) = \begin{cases} R_i(t)^\mu \\ -\lambda(-R_i(t))^\nu, \end{cases} \tag{3}$$

where $R_i(t)$ is the reward associated with option i at trial t, and $\alpha^\pm$ indicates the learning rate. $U(R_i(t))$ takes the form of the prospect utility function proposed by Tversky and Kahneman [30] in which μ and ν measure the degrees of risk aversion and risk seeking, respectively. We adopted this utility function because one of our research interests is the effect of risk attitudes on creativity. Thus, it was assumed that participants would evaluate the reward in terms of their own risk attitudes, which resulted in the utility function specified in [3].

The reward prediction error $\delta(t)$ is computed by subtracting the current value estimate from the obtained reward R. Participants thus update the action value estimate by scaling the prediction error with the learning rate and then adding this to the estimated value at the previous trial. Learning rates close to 1 indicate that a person makes fast adaptations based on prediction errors, and learning rates closer to 0 indicate slow adaptation. In the default setting, the initial action values are set to zero so that $Q_i(1) = 0$ for i = 1,...,4.

For the unchosen option j (i≠j), the action value is updated as:

$$Q_j(t+1) = Q_j(t). \tag{4}$$

Assume the chosen action at trial t is denoted by a(t)∈{1,2,3,4}. The action value estimates of these four options are used to determine the probability to choose either option. This probability is computed via the following softmax decision rule:

$$P(a(t) = i) = \frac{exp(\beta Q_i(t))}{\sum_{j=1}^{4} exp(\beta Q_j(t))}, \tag{5}$$

where $P(a(t) = i)$ is the probability to choose the action a(t) = i at trial t. The parameter β is the inverse temperature, a parameter that indicates the sensitivity of a participant's choice to the difference in action value estimates. As described below, the inverse temperature β measures the relative strength of exploitation vs. exploration.

The parameters of α and β in this model were estimated by optimizing the maximum a posteriori (MAP) objective function, that is, finding the posterior mode:

$$\hat{\theta} = argmax \, p(D_s|\theta_s)\mathrm{p}(\theta_s), \tag{6}$$

where $p(D_s|\theta_s)$ is the likelihood of data $D_s$ for subject s conditional on parameters $\theta_s = \{\alpha^S, \beta^S\}$, and $\mathrm{p}(\theta_s)$ is the prior probability of $\theta_s$. We assume each parameter is bounded and use constrained optimization to find the MAP estimates. More specifically, since α is bounded between 0 and 1 and β takes non-negative values, their priors were assumed to follow beta distributions for α, and gamma distributions for β.

Although this Q learning model was specified for the IGT, we believe this model could also account for crucial aspects of creative thinking. In creative thinking, some ideas or alternatives must be chosen to further scrutinize existing possibilities. For this purpose, the individual must evaluate each idea or alternative before choosing one option. This aspect of decision making in creative thinking could be regarded as exercising the above Q learning model. In particular, convergent thinking hinges on this type of decision making to arrive at the correct solution. Even in divergent thinking, idea exploration requires the preliminary evaluation of candidate search fields. Therefore, the Q learning model could be interpreted as a model of creative thinking as well as a model for describing behavior in the IGT.

## 2.3. Measures

This paper used divergent and convergent thinking scores as dependent variables in the regression analysis. This is because there is growing support for taking both divergent and convergent thinking into account in creativity [31]. Our primary interest lies in the effects of exploitation and exploration on these dependent variables. As control variables that might affect divergent and convergent thinking, we used working memory capacities and personality scales.

**2.3.1. Divergent thinking.** Divergent thinking is defined as the ability to produce new approaches and original ideas by forming unexpected combinations from available information, and by applying such abilities as semantic flexibility, and fluency of association, ideation, and transformation [32]. In the current study, divergent thinking ability was measured with the S-A creativity test [33], a timed laboratory test corresponding to the measures used in the Torrance Test of Creative Thinking. The test involves three types of tasks. In the first task, participants are instructed to generate alternative ways of using objects specified in the test, which is known as an 'alternative use' test (AUT). The second task requires the participant to imagine desirable (and not realized) functions of specified ordinary objects. In the third task, participants are instructed to imagine the consequences of "unimaginable things" happening. Each task requires participants to generate as many answers as possible (up to 10).

The S-A creativity test measures divergent thinking in terms of (a) fluency, (b) flexibility, (c) originality, and (d) elaboration. Fluency is measured by the number of relevant responses to the questions and is related to the ability to produce and consider many alternatives. Flexibility is the ability to produce responses from a broad perspective and is measured by the sum of the total number of category types that answers are assigned based on a criteria table or an almost equivalent judgment. Originality is the ability to produce ideas that differ from others and its scoring is based on the sum of the idea categories that are weighted based on a criteria table or an almost equivalent judgment. Elaboration is the ability to produce ideas in detail and is measured by the sum of answers that are weighted based on a criteria table or an almost equivalent judgment. This test also provides the total score for divergent thinking, which was mainly used in this paper. For more detail about the S-A creativity test, see [34].

**2.3.2. Convergent thinking.** Convergent thinking is defined as the ability to apply conventional and logical search, recognition, and decision-making strategies to stored information to produce an already known answer [35]. Thus, convergent thinking requires knowledge and is typically correlated with measures of crystallized intelligence. However, most creativity researchers have described convergent thinking as a process entailing evaluation of initial ideas and/or a sudden insight in arriving at the correct solution for problems with task constraints [35–37]. As a result, in the insight problem-solving literature, convergent thinking has been typically measured by the Remote Associates Test (RAT) [38], which entails task constraints that the correct solution must fit with each of the three words in the presented triad (e.g., cheese for 'cottage, cream, blue'). As all the participants in this study were native Japanese, we adopted a Japanese version of RAT developed by [39] where words were represented by Chinese characters. We used 40 problems selected by [40] in our experiment. RAT (convergent thinking) scores were measured by the number of correct solutions for the 40 problems.

**2.3.3. Inverse temperature.** We used the inverse temperature β to represent levels of exploitation and exploration. A higher β value implies that participants selected the decks based on the action value Q calculated in (1)~(3), leading to exploitation. Conversely, as β approaches zero, the choice is more likely to have been made randomly because the weight of the Q value in the soft max decision rule in (3) significantly declines. This implies that participants undertake exploration. Thus, the inverse temperature β measures the relative importance of exploitation and exploration.

**2.3.4. Risk attitudes.** As describe above, risk attitudes can be measured by the parameters μ and ν in (3), which incorporate part of the prospect utility function in which an asymmetric form of risk aversion is specified. Risk aversion in cases with positive rewards and risk seeking (risk-taking) in cases with negative rewards are respectively measured by μ and ν, which reflects the idea that participants have different risk attitudes towards gains and losses. We were interested in examining these effects on creativity performance.

**2.3.5. Working Memory Capacity (WMC).** WMC was measured using reading span, operation span, and matrix span tests, which are representative working memory tests [41]. Reading span and operation span tests evaluate the capacity of verbal WMC and logical WMC, respectively, which in turn correspond to the phonological loop, according to Baddeley's model [42]. The matrix span test measures spatial WMC, corresponding to the visuo-spatial sketchpad in this model.

**2.3.6. Big five scale of personality.** Big Five Scales (BFS) of personality traits are widely used to describe personality differences, which consist of five factors, namely openness to experience (inventive/curious vs. consistent/cautious), conscientiousness (efficient/organized vs. easy-going/careless), extraversion (outgoing/energetic vs. solitary/reserved), agreeableness (friendly/compassionate vs. challenging/detached), and neuroticism (sensitive/nervous vs. secure/confident) [43–45]. These scales were measured by 60 questions in Japanese, developed by [46]. The scores were measured in descending order. For example, high scores in openness to experience imply lower openness to experience. The descriptive statistics for all these variables used in the empirical analyses in this paper are reported in Table 1.

## 2.4. Procedure

Participants completed the S-A, Japanese RAT, reading span, operation span, and matrix span tests, the IGT and BFS tests. The experiments were presented in two independent sessions: a S-A/RAT session and a WMC/IGT/BFS session. Approximately half of the participants performed the two sessions in the order of the S-A/RAT session and then the WMC/IGT/BFS

**Table 1. Descriptive statistics.**

| | Mean | SD | 1 | 2 | 3 | 4 | 5 | 6 | 7 | 8 | 9 | 10 | 11 | 12 |
|---|---|---|---|---|---|---|---|---|---|---|---|---|---|---|
| **Descriptive statistics** | | | | | | | | | | | | | | |
| 1. Divergent Thinking | 39.87 | 10.07 | — | | | | | | | | | | | |
| 2. Convergent Thinking | 14.49 | 3.94 | -0.08 | — | | | | | | | | | | |
| 3. μ (risk aversion in gains) | 0.24 | 0.34 | -0.08 | 0.07 | — | | | | | | | | | |
| 4. ν (risk-seeking in losses) | 0.26 | 0.34 | 0.09 | 0.03 | 0.52*** | — | | | | | | | | |
| 5. Inverse temperature | 1.36 | 1.88 | 0.04 | -0.02 | 0.59*** | 0.66*** | — | | | | | | | |
| 6. Extraversion | 3.61 | 0.95 | -0.05 | 0.12 | -0.01 | 0.13 | 0.07 | — | | | | | | |
| 7. Neuroticism | 3.29 | 0.97 | 0.08 | 0.02 | -0.07 | 0.00 | -0.02 | -0.40*** | — | | | | | |
| 8. Openness | 3.89 | 0.83 | -0.16* | 0.00 | -0.05 | 0.04 | -0.09 | 0.23*** | -0.21** | — | | | | |
| 9. Conscientiousness | 4.19 | 0.69 | 0.00 | 0.20* | 0.04 | 0.14 | 0.08 | 0.08 | 0.10 | 0.04 | — | | | |
| 10. Agreeableness | 3.4 | 0.89 | 0.12 | -0.08 | -0.25*** | -0.18** | -0.14 | 0.37*** | -0.24*** | 0.14 | 0.29*** | — | | |
| 11. Spatial WMC | 23.81 | 13.38 | 0.13 | 0.16* | -0.08 | 0.00 | 0.04 | 0.04 | 0.00 | -0.10 | 0.01 | 0.02 | — | |
| 12. Verbal WMC | 25.73 | 12.75 | 0.00 | 0.07 | 0.06 | 0.05 | 0.05 | 0.05 | -0.07 | -0.02 | 0.01 | -0.08 | 0.37*** | — |
| 13. Logical WMC | 28.18 | 11.65 | 0.05 | -0.12 | 0.02 | 0.11 | 0.10 | -0.02 | 0.01 | 0.07 | -0.01 | -0.02 | 0.22** | 0.24*** |

*$p < .10$

**$p < .05$

***$p < .0.01$

session. The remaining participants performed the sessions in the opposite order. There was at least a 10-day interval between the two successive sessions.

During the S-A/RAT session, participants completed the S-A test and RAT, which took approximately 70 minutes. The S-A test was completed in accordance with the instructor's manuals. In the RAT, participants were given a practice session using two examples. Following the practice session, they engaged in 40 problems. Each problem was presented for 40 seconds to the participants. After 20 problems, a one-minute break was taken. Problems were presented in a randomized order each time, to remove order effects. A break of at least 5 minutes was taken between the two tests.

In the WMC/IGT/BFS session, participants completed reading span, operation span, matrix span tests [41], IGT, and BFS tests, which took approximately 60 minutes. This session was arranged for groups with a maximum of 50 participants in the presence of the instructor. The tests were performed on a 17" CRT monitor with PsytoolKit [47, 48]. A break of at least 1 minute was taken between the three tests. The order of these tests was also randomly assigned in this session. In the following discussion, reading span, operation span, and matrix span test scores are denoted by verbal WMC, logical WMC, and spatial WMC, respectively.

## 3. Results

### 3.1. Learning convergence

First, we analyzed the data from each participant regarding learning convergence. Each participant completed 100 trials of the IGT, and some of them eventually learned to consistently select the best (low risk, low return) decks. When participants repeatedly chose the best decks at least four times consecutively until the end of the game, we classified these participants as those in which learning convergence took place. Based on this definition, we identified 60 participants who succeeded in achieving learning convergence. In these participants, the average

number of trials that were completed before learning convergence took place was 70.9 (SD = 29.5). To avoid overestimation of exploitation for those participants, we excluded the data from trials that took place after the achievement of learning convergence from the subsequent analysis. Thus, in addition to the pooled sample, we classified two subsamples, successful and unsuccessful participants, in terms of learning convergence.

## 3.2. Determinants of divergent thinking

To examine the effects of risk attitudes, exploitation and exploration on divergent and convergent thinking in the Q learning model, we first estimated the parameters of the learning rate (α), inverse temperature (β), risk aversion index for gains (μ), and risk seeking index for losses (ν) through the MAP estimation described above. We then implemented regression analyses on the determinants of both divergent and convergent thinking scores. Because both divergent and convergent thinking scores take only nonnegative integer values, we used the Poisson regression method to maintain statistical consistency.

First, we examined the data with respect to divergent thinking. The results are reported in Table 2. The dependent variable was the divergent thinking score. Columns (1) and (2) in the table show the results for the pooled sample, columns (3) and (4) refer to the successful subsample, and columns (5) and (6) show the results for the unsuccessful subsample.

In the pooled sample, the risk aversion index μ exerted a negative effect whereas the risk seeking index ν had a positive effect on divergent thinking. This implies that participants with learning convergence behaved in a risk-seeking manner, regardless of the gains or losses. Conversely, the inverse temperature β had no effect on divergent thinking, which suggests that risk attitudes, rather than exploitation/exploration, accounted for high levels of divergent thinking.

These results were also retained for the successful subsample. Regarding the control variables, openness to experience, extroversion, and agreeableness were identified as determinants of divergent thinking. Except for spatial WMC, the results were similar to those in the pooled sample. Since personality variables were measured in descending order, both openness to experience and extraversion exerted positive effects on divergent thinking. Moreover, the negative effect of agreeableness suggests that highly divergent thinkers favor challenging, rather than accepting, existing ideas. These results seem consistent with the definition of divergent thinking.

Regarding unsuccessful participants, the risk aversion index μ was negatively related to divergent thinking, whereas inverse temperature, neuroticism, and spatial WMC had positive contributions. The effect of the risk aversion μ was the same as that in successful divergent thinkers, which implies that both successful and unsuccessful divergent thinkers showed risk seeking in the face of positive gains. However, unsuccessful divergent thinkers relied more heavily on the results of the Q values due to the positive effect of inverse temperature. Compared with successful divergent thinkers, this exploitation effect seemed to inhibit learning convergence, although these participants achieved high divergent thinking scores. The positive effect of neuroticism likely contributed to high divergent thinking scores because high sensitivity facilitated internal memory search during the S-A test.

As Akbari, Chermahini, & Hommel [49, 50] reported nonlinear relationships between creativity and some of its determinants, we also tested the nonlinear effects of risk attitudes and inverse temperature in columns (2), (4), and (6) in Table 2. The positive effects of ν squared were identified according to columns (2) and (4), and the opposite effects were found according to column (6). Thus, in the pooled and successful samples, increasing returns emerged with respect to ν, which indicates the strong effect of ν on divergent thinking. In unsuccessful

**Table 2. Determinants of divergent and convergent thinking.**

| Variables | Determinants of divergent and convergent thinking | | | | | | | | | | |
| | (SE in parentheses) | | | | | | | | | | |
| | Divergent thinking | | | | | | Convergent thinking | | | | |
| | Pooled | | Success | | No success | | | Pooled | | Success | | No success | |
| | (1) | | (2) | | (3) | | | (4) | | (5) | | (6) | |
| Constant Terms | 43.46 | *** | 65.89 | *** | 20.98 | * | | 7.89 | * | 13.74 | * | 5.23 | |
| | (6.19) | | (9.44) | | (9.24) | | | (3.66) | | (5.42) | | (5.45) | |
| μ (risk aversion in gains) | -3.97 | . | -5.73 | . | -5.91 | * | | 1.52 | | -1.46 | | -1.94 | |
| | (2.30) | | (3.01) | | (2.63) | | | (1.42) | | (1.83) | | (1.57) | |
| ν (risk-seeking in losses) | 7.09 | ** | 11.10 | ** | -1.52 | | | -0.59 | | -3.40 | | 1.61 | |
| | (2.52) | | (3.40) | | (3.24) | | | (1.53) | | (2.14) | | (1.94) | |
| Inverse temperature | -0.12 | | -0.29 | | 1.24 | * | | -0.24 | | -0.15 | | 0.56 | . |
| | (0.46) | | (0.51) | | (0.52) | | | (0.28) | | (0.30) | | (0.31) | |
| Exraversion | -1.00 | | -3.21 | ** | 1.12 | | | 0.87 | . | 0.89 | | 0.93 | |
| | (0.75) | | (1.10) | | (1.09) | | | (0.45) | | (0.65) | | (0.65) | |
| Neuroticism | 0.49 | | -1.26 | | 2.10 | * | | 0.14 | | -0.40 | | 0.56 | |
| | (0.71) | | (1.06) | | (1.02) | | | (0.43) | | (0.63) | | (0.61) | |
| Openness | -1.98 | ** | -5.65 | *** | -0.19 | | | 0.09 | | 1.99 | * | -0.93 | |
| | (0.76) | | (1.26) | | (1.05) | | | (0.46) | | (0.78) | | (0.62) | |
| Conscientiousness | -1.26 | | -0.09 | | -0.43 | | | 1.44 | * | 1.27 | | 1.77 | * |
| | (0.94) | | (1.55) | | (1.26) | | | (0.56) | | (0.95) | | (0.75) | |
| Agreeableness | 2.49 | ** | 3.90 | ** | 1.61 | | | -0.98 | * | -2.36 | ** | -0.65 | |
| | (0.82) | | (1.37) | | (1.05) | | | (0.49) | | (0.84) | | (0.63) | |
| Spatial WMC | 0.09 | . | -0.08 | | 0.22 | ** | | 0.06 | * | 0.05 | | 0.12 | * |
| | (0.05) | | (0.07) | | (0.08) | | | (0.03) | | (0.04) | | (0.05) | |
| Verbal WMC | -0.03 | | 0.06 | | -0.05 | | | 0.01 | | -0.01 | | -0.02 | |
| | (0.05) | | (0.08) | | (0.07) | | | (0.03) | | (0.05) | | (0.04) | |
| Logical WMC | 0.03 | | -0.07 | | 0.06 | | | -0.05 | . | -0.15 | ** | -0.03 | |
| | (0.05) | | (0.08) | | (0.08) | | | (0.03) | | (0.05) | | (0.05) | |
| AIC | 905.31 | | 432.13 | | 460.71 | | | 634.76 | | 305.19 | | 330.04 | |

*$p < .10$
**$p < .05$
***$p < .01$

divergent thinkers, however, this effect had an inverted U shape, which suggests a much weaker effect.

Despite these contrasting effects, the results indicated that both successful and unsuccessful divergent thinkers engaged in risk seeking in the face of losses. Overall, highly divergent thinkers were more likely to engage in risk seeking as opposed to risk aversion, and the effects of exploitation and exploration were mostly negligible.

## 3.3. Determinants of convergent thinking

Next, we examine the determinants of convergent thinking. The results are reported in Table 3. The dependent variable was the convergent thinking score. As in Table 2, columns (1) and (2) in the table show the results for the pooled sample, columns (3) and (4) refer to the successful subsample, and columns (5) and (6) show the results for the unsuccessful subsample.

Table 3. Determinants of divergent and convergent thinking.

| Variables | Divergent thinking | | | | | Convergent thinking | | | | | |
|---|---|---|---|---|---|---|---|---|---|---|---|
| | Pooled | | Success | | No success | | Pooled | | Success | | No success | |
| | (1) | | (2) | | (3) | | (4) | | (5) | | (6) | |
| Constant Terms | 43.42 | *** | 67.87 | *** | 12.51 | | 7.47 | * | 13.91 | * | 4.52 | |
| | (6.21) | | (9.56) | | (10.29) | | (3.66) | | (5.46) | | (6.08) | |
| $\mu$ (risk aversion in gains) | 2.71 | | -3.47 | | -9.80 | | -0.54 | | -2.32 | | 2.17 | |
| | (10.28) | | (11.79) | | (11.44) | | (5.96) | | (6.94) | | (6.84) | |
| $\mu^2$ (risk aversion in gains) | -5.94 | | -3.23 | | 4.03 | | 2.23 | | 0.50 | | -4.25 | |
| | (10.47) | | (11.54) | | (11.41) | | (6.22) | | (6.96) | | (6.84) | |
| $\nu$ (risk-seeking in losses) | -13.07 | | -12.16 | | 23.33 | . | 7.76 | | -1.81 | | 2.92 | |
| | (10.59) | | (12.33) | | (13.14) | | (6.45) | | (7.46) | | (7.90) | |
| $\nu^2$ (risk-seeking in losses) | 20.63 | * | 21.78 | * | -25.29 | . | -7.60 | | -1.75 | | -1.12 | |
| | (10.01) | | (11.02) | | (13.12) | | (6.07) | | (6.59) | | (7.89) | |
| $\beta$ (Inverse temperature) | 0.32 | | 0.14 | | 2.16 | | -1.24 | | -1.73 | . | 0.79 | |
| | (1.39) | | (1.72) | | (1.36) | | (0.82) | | (1.05) | | (0.82) | |
| $\beta^2$ (Inverse temperature) | 0.00 | | -0.01 | | -0.09 | | 0.12 | | 0.20 | | -0.02 | |
| | (0.18) | | (0.22) | | (0.15) | | (0.11) | | (0.13) | | (0.09) | |
| Exraversion | -1.04 | | -3.23 | ** | 1.57 | | 0.96 | * | 1.35 | . | 0.95 | |
| | (0.76) | | (1.16) | | (1.14) | | (0.45) | | (0.71) | | (0.67) | |
| Neuroticism | 0.44 | | -1.49 | | 2.35 | * | 0.25 | | -0.14 | | 0.52 | |
| | (0.72) | | (1.09) | | (1.05) | | (0.43) | | (0.65) | | (0.62) | |
| Openness | -1.93 | * | -5.30 | *** | 0.02 | | 0.03 | | 2.00 | * | -1.01 | |
| | (0.77) | | (1.28) | | (1.08) | | (0.47) | | (0.79) | | (0.65) | |
| Conscientiousness | -1.17 | | 0.20 | | -0.40 | | 1.41 | * | 1.30 | | 1.63 | * |
| | (0.94) | | (1.54) | | (1.32) | | (0.56) | | (0.95) | | (0.78) | |
| Agreeableness | 2.31 | ** | 3.52 | * | 1.63 | | -0.98 | . | -2.55 | ** | -0.53 | |
| | (0.83) | | (1.43) | | (1.09) | | (0.50) | | (0.87) | | (0.65) | |
| Spatial WMC | 0.08 | . | -0.08 | | 0.27 | ** | 0.07 | * | 0.07 | | 0.12 | * |
| | (0.05) | | (0.08) | | (0.08) | | (0.03) | | (0.04) | | (0.05) | |
| Verbal WMC | -0.01 | | 0.08 | | -0.09 | | 0.00 | | -0.01 | | -0.03 | |
| | (0.05) | | (0.08) | | (0.07) | | (0.03) | | (0.05) | | (0.04) | |
| Logical WMC | 0.04 | | -0.06 | | 0.06 | | -0.06 | . | -0.15 | ** | -0.03 | |
| | (0.06) | | (0.08) | | (0.08) | | (0.03) | | (0.05) | | (0.05) | |
| AIC | 906.71 | | 305.19 | | 462.25 | | 638.32 | | 308.47 | | 335.52 | |

$^*p < .10$

$^{**}p < .05$

$^{***}p < .0.01$

According to this table, neither successful nor unsuccessful divergent thinking were associated with risk attitudes in any columns. However, inverse temperature was weakly correlated with unsuccessful divergent thinking. Therefore, regarding convergent thinking measured by the RAT, risk attitudes and exploitation/exploration did not play significant roles in achieving higher performance.

Regarding control variables, openness to experience, agreeableness, and logical WMC were identified in the successful subsample in both columns (3) and (4). In contrast to divergent thinking, successful convergent thinking hinged on negative openness to experience.

Moreover, agreeableness was now positively related to convergent thinking, which implies that accepting, rather than challenging, the result of a memory search led to higher convergent thinking scores. Regarding the negative effects of logical WMC on convergent thinking, it is possible that logic per se does not play a critical role in the RAT because searching for the correct Chinese character that completes three words simultaneously requires more trial and error learning than logical reasoning. The results appear to suggest that high logical WMC impedes this trial and error learning attempt in the RAT in favor of logical reasoning.

As for unsuccessful convergent thinkers, conscientiousness and spatial WMC were significant in both columns (5) and (6). Once again, because the RAT assumed correct solutions, several candidate solutions must be organized efficiently to come up with correct solutions. As for spatial WMC, our RAT experiment used hieroglyphic Chinese characters, suggesting that the spatial WMC is positively related to convergent thinking. The results in the pooled sample reflected those results in an eclectic way.

Although the importance of these control variables could be interpreted intuitively, given our research interests, we want to emphasize our findings that that risk attitudes and exploitation/exploration did not play significant roles in convergent thinking for both the successful and unsuccessful subsamples.

## 4. Discussion

In this paper, the underlying cognitive mechanism of creativity was modeled in the RL framework. Recent computational analyses based on the RL framework were applied to studies of decision making and learning in various social contexts [7–12], showing the close correspondence between the phasic dopaminergic firing and the reward prediction error [6]. Although Hills, Todd, & Goldstone [51] made the first attempt to relate exploitation and exploration to creativity tasks, and several studies have directly modeled the creativity process (see [52] for review), we propose an alternative computational model of creativity using the simple RL framework. One of the contributions of this paper was that we examined the effects of risk attitudes and exploitation/exploration on creativity performance using this RL framework.

From this perspective, the current study revealed the novel finding that divergent thinking was explicitly related to risk seeking, whereas convergent thinking was not associated with risk attitudes or exploitation/exploration. These results stand in sharp contrast to the findings reported by Shen, Hommel, Yuan, Chang, & Zhang [23], who found that while low risk-taking contributed to better convergent thinking, it was not significantly correlated with divergent thinking. One reason for this disparity could be the different measures of risk attitudes. Namely, Shen, Hommel, Yuan, Chang, & Zhang [23] used the risk-taking preference index, while in the current study, we measured risk attitudes by imposing a prospective utility function during the IGT. These different measures could highlight separate aspects of risk-taking. The differences between the risk-taking preference index and the risk attitudes derived from the utility function should be explored in future studies.

Although they were not identical, Tyagi, Hanoch, Hall, Runco, & Denham [22] employed similar risk-taking measures to Shen, Hommel, Yuan, Chang, & Zhang [23]. They found that creativity was associated with high risk-taking tendencies in the social domain, and that the likelihood of social risk taking was the strongest predictor of creative personality and ideation scores. These results are consistent with our present findings.

The contrasting results among different studies might reflect cultural differences. As Shen, Hommel, Yuan, Chang, & Zhang [23] noted, while divergent and convergent thinking were positively correlated in their study, the experiment conducted in the Netherlands revealed correlations that were close to zero and, if anything, negative (e.g., [49]). In our study, as revealed

in Table 1, the correlation between divergent and convergent thinking was close to zero. Thus, different roles during risk-taking might reflect cultural differences between China and Japan.

Regarding convergent thinking, it was found that both risk attitudes and inverse temperature did not account for its performance at least explicitly. However, we should be cautious about interpreting this result because, as we have noted, the RAT required both divergent and convergent thinking to yield correct solutions. Moreover, the correct solution was sometimes obtained through insight. That is, some participants solved some questions with insight. The mixture of problem solving with and without insight makes it difficult to identify the relative contributions of exploitation and exploration and risk attitudes. It has been claimed that solutions to insight problems are unpredictable [53], difficult to report [54], and solved in a distinct manner [55]. In other words, insight generates new information that is often discrete and domain-specific and transcends informational boundaries, yet still provides some value. This might be the result of underlying risk-taking exploration activities. In contrast, problem solving without insight goes through a step-by-step, risk averse incremental process, which corresponds to exploitation. Consequently, it could be that the mixture of problem solving with and without insight in the RAT obscures significant effects of both exploitation and exploration.

Insight problem solving does demand distant or remote associations, such as spreading activation in semantic memory, which operates largely outside of conscious attentional control [27, 56–58]. This suggests that it requires substantially less working memory than problem solving without insight. In support of this, it has been shown that higher WMC is positively related to analytical problem solving, but unrelated to insight [59]. Similarly, a higher degree of working memory load negatively affects analytic problem solving but has no impact on insight problem solving [55].

Problem solving with and without insight are likely to both involve many of the same cognitive processes and neural mechanisms. Indeed, some studies have suggested that insight plays no role and that creativity is essentially identical to analytical problem solving [56, 60–63]. Murray and Byrne [64] suggested that people with high WMC perform better on insight problems than people with less WMC. De Dreu, Nijstad, Baas, Wolsink, and Roskes [65] also performed several studies that indicated that high WMC was related to higher performance in a range of different creative tasks. Because both insight and analytical problem solving were involved in the RAT, these studies were consistent with our results that the spatial WMC was positively related to higher convergent thinking scores for unsuccessful participants.

The reason that only the spatial WMC positively accounted for convergent thinking was that the RAT used hieroglyphic Chinese characters in our experiment. Participants were required to memorize the forms of Chinese characters while searching for solutions. Regarding the negative effect of logical WMC on convergent thinking, as noted above, it is possible that logic per se does not play a critical role in the RAT because it requires trial and error learning rather than logical reasoning. The current results appear to suggest that high logical WMC impedes this trial and error approach in the RAT in favor of logical reasoning.

Spatial WMC also positively accounted for divergent thinking, except for in successful participants. In the alternative use test that was part of the S-A test, participants were asked to write alternative uses of given items. In this test, participants had to keep track of images of given items. Those who were slower learners might have needed to keep the image in their memory for longer, requiring higher spatial WMC. However, quick learners would not have to keep the image in their memory for long, meaning less reliance on spatial WMC.

The different effects of risk-taking on divergent thinking between successful and unsuccessful participants also deserve some remarks here. While risk seeking in the face of gains was positively related to divergent thinking for both subsamples, risk-seeking in the face of losses accounted for divergent thinking only for quick learners (successful participants). This result

suggests that quick learners were more sensitive to losses, leading to more risk-taking behaviors. Thus, greater risk-taking regarding losses might differentiate quick from slow learners.

It should also be noted that inverse temperature (exploitation/exploration) did not account for both divergent and convergent thinking. Although we did not report in this paper, regression analyses without risk parameters revealed that inverse temperature was strongly significant for divergent thinking. However, once the risk parameters were incorporated, the significance was lost. This implies that risk attitudes, rather than exploitation/exploration, accounted for divergent thinking in our model. Therefore, as related studies have suggested [22, 23], risk attitudes appear to play a critical role in creativity.

Thus, the current findings revealed that divergent thinking tends to favor risk-taking rather than exploitation and exploration. In this respect, related neuroscientific studies of creativity [2, 66, 67] have underlined the importance of the default mode network in divergent thinking [68–72]. However, while the identification of neural substrates underlying divergent thinking has certainly improved our understanding of creativity mechanisms, the specific cognitive model behind the divergent thinking process has not yet been rigorously articulated.

This paper contributes to linking risk-taking to divergent thinking in the framework of Q learning and empirically confirms their relationship. However, the challenge of articulating how risk-taking in divergent thinking is related to underlying neural substrates, particularly the default mode network, remains unresolved.

Finally, it should be pointed out that risk parameters, exploitation and exploration in this study were not measured during divergent/convergent thinking tasks. Therefore, the relationships exhibited in the current study between risk attitudes, exploitation/exploration, and divergent/convergent thinking should be considered as indirect evidence, and the results do not unequivocally confirm that divergent thinkers always implement risk-taking behaviors. Obviously, measuring risk attitudes, exploitation and exploration during creative tasks remains a challenge for future research.

## 5. Conclusion

This paper related risk attitudes, exploitation and exploration specified in a simple Q learning model to creativity measured by convergent and divergent thinking. In other words, we examined the determinants of divergent and convergent thinking in terms of risk attitudes, exploitation and exploration as cognitive processes. While many creativity studies focus on neural substrates of creativity using EEG and fMRI data, this study was different in its introduction of a computational approach to creativity research.

Findings revealed novel characteristics of risk-taking in divergent thinking. That is, efficient divergent thinkers tend to engage in risk-taking, rather than risk-averting behavior. In particular, quick learners are more likely to exhibit higher levels of risk-taking in the face of losses compared with gains. This risk-taking behavior not only contributes to fast learning in the IGT, but also facilitates high performance in terms of divergent thinking.

To our knowledge, this study is the first to explore the cognitive mechanisms underlying creativity in the framework of reinforcement learning. Future progress in this area will require several challenges to be addressed. In particular, neural substrates of both risk attitudes and exploitation/exploration, together with convergent and divergent thinking, should be identified with neuroscientific methods such as EEG and fMRI.

## Author Contributions

**Conceptualization:** Tsutomu Harada.

**Data curation:** Tsutomu Harada.

**Formal analysis:** Tsutomu Harada.

**Funding acquisition:** Tsutomu Harada.

**Investigation:** Tsutomu Harada.

**Methodology:** Tsutomu Harada.

**Project administration:** Tsutomu Harada.

**Resources:** Tsutomu Harada.

**Software:** Tsutomu Harada.

**Supervision:** Tsutomu Harada.

**Validation:** Tsutomu Harada.

**Visualization:** Tsutomu Harada.

**Writing – original draft:** Tsutomu Harada.

**Writing – review & editing:** Tsutomu Harada.

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
