## [Decision Letter · Decision Letter 0]

16 Mar 2020

PONE-D-20-02635

Effects of Exploitation vs. Exploration on Creativity in Q Learning

PLOS ONE

Dear Prof. Harada,

Thank you for submitting your manuscript to PLOS ONE. After careful consideration, we feel that it has merit but does not fully meet PLOS ONE’s publication criteria as it currently stands. Therefore, we invite you to submit a revised version of the manuscript that addresses the points raised during the review process.

Though Reviewer 1 rejected PONE-D-20-02635, the reviewer provided many valuable and constructive comments. Considering both reviewers’ useful comments and the interesting topic of the manuscript, I would like to give you a chance to revise your manuscript during the special period. The revised manuscript will undergo the next round of review by the same reviewers. 

We would appreciate receiving your revised manuscript by Apr 30 2020 11:59PM. To enhance the reproducibility of your results, we recommend that if applicable you deposit your laboratory protocols in protocols.io, where a protocol can be assigned its own identifier (DOI) such that it can be cited independently in the future. For instructions see: http://journals.plos.org/plosone/s/submission-guidelines#loc-laboratory-protocols

We look forward to receiving your revised manuscript.

Kind regards,

Baogui Xin, Ph.D.

Academic Editor

PLOS ONE

Journal Requirements:

2. Please change "female” or "male" to "woman” or "man" as appropriate, when used as a noun.

Reviewers' comments:

Reviewer's Responses to Questions

**Comments to the Author**

1. Is the manuscript technically sound, and do the data support the conclusions?

Reviewer #1: No

Reviewer #2: Partly

Reviewer #3: No

2. Has the statistical analysis been performed appropriately and rigorously? 

Reviewer #1: No

Reviewer #2: No

Reviewer #3: N/A

3. Have the authors made all data underlying the findings in their manuscript fully available?

Reviewer #1: Yes

Reviewer #2: Yes

Reviewer #3: Yes

4. Is the manuscript presented in an intelligible fashion and written in standard English?

Reviewer #1: Yes

Reviewer #2: Yes

Reviewer #3: Yes

5. Review Comments to the Author

Reviewer #1: The manuscript addresses the question of how two measures of creativity (divergent vs. convergent) thinking relates to exploration – exploitation strategies in the version of Iowa Gambling Task. The author made a prediction that divergent thinking measured through the S-A creativity test is related to more exploratory behavior while convergent thinking should bias behavioral strategies towards more exploitation. However, the obtained results do not confirm this hypothesis. Individuals who scored more on divergent thinking show some relationship to both exploration and exploitation (which depends on the chosen Q-learning model) while convergent thinking does not show any relationship with exploration-exploitation trade-off in this particular task.The author then made some post-hoc interpretations trying to reconcile the obtained results. Despite the fact that I appreciate the effort of trying to investigate the link between such a complex phenomenon as creativity and learning, these results alone do not lead to any conclusion and do not provide any clear guidelines to future research and, hence, could not be published.

Reviewer #2: The approach underlying this study is very interesting and the general method is useful. However, some possibly crucial information is missing, which may have serious impact on the results and their interpretation. The paper also shows a lack of scholarship, as it is presented as the first to consider the difference between convergent and divergent thinking--thus ignoring numerous studies looking into the difference between these two types of creativity. All this makes it difficult to judge whether the paper makes a useful contribution to the literature.

MAJOR

1. A big deal is made of the difference between divergent and convergent thinking. Not only does this ignore the key point of Guilford, who had introduced this terminological difference, but it also overlooks quite a number of studies comparing divergent and convergent thinking. For convenience, I list just a number of papers from my own lab:

Prochazkova, L., Lippelt, D.P., Colzato, L.S., Kuchar, M., Sjoerds, Z., & Hommel, B. (2018). Exploring the effect of microdosing psychedelics on creativity in an open-label natural setting. Psychopharmacology, 235, 3401-3413.

Shen, W., Hommel, B., Yuan, Y., Chang, L. & Zhang, W. (2018). Risk-taking and creativity: Convergent, but not divergent thinking is better in low-risk takers. Creativity Research Journal, 30, 224-231.

Colzato, L.S., de Haan, A., & Hommel, B. (2015). Food for creativity: Tyrosine promotes deep thinking. Psychological Research, 79, 709-714.

Zmigrod, S., Colzato, L.S., & Hommel, B. (2015). Stimulating creativity: modulation of convergent and divergent thinking by tDCS. Creativity Research Journal, 27, 353-360.

Zmigrod, S., Zmigrod, L., & Hommel, B. (2015). Zooming into creativity: Individual differences in attentional global-local biases are linked to creative thinking. Frontiers in Psychology, 6:1647.

Lippelt, D.P., Hommel, B., & Colzato, L.S. (2014). Focused attention, open monitoring and loving kindness meditation: Effects on attention, conflict monitoring and creativity. Frontiers in Psychology, 5:1083.

Reedijk, S.A., Bolders, A., & Hommel, B. (2013). The impact of binaural beats on creativity. Frontiers in Human Neuroscience, 7:786.

Hommel, B., Colzato, L.S., Fischer, R., & Christoffels, I. (2011). Bilingualism and creativity: Benefits in convergent thinking come with losses in divergent thinking. Frontiers in Psychology, 2:273.

Akbari Chermahini, S., & Hommel, B. (2010). The (b)link between creativity and dopamine: Spontaneous eye blink rates predict and dissociate divergent and convergent thinking. Cognition, 115, 458-465.

Hommel, B. (2012). Convergent and divergent operations in cognitive search. In: P.M. Todd, T.T. Hills, & T.W. Robbins (eds.), Cognitive search: Evolution, algorithms, and the brain (pp. 221-235). Cambridge, MA: MIT Press.

2. The author also ignores existing attempts to specify the mechanisms underlying convergent and divergent thinking; e.g.,:

Zhang, W., Sjoerds, Z., & Hommel, B. (2020). Metacontrol of human creativity: The neurocognitive mechanisms of convergent and divergent thinking. Neuroimage, 210:116572.

Mekern, V., Hommel, B., & Sjoerds, Z. (2019). Computational models of creativity: A review of single- and multi-process recent approaches to demystify creative cognition. Current Opinion in Behavioral Sciences, 27, 47-54.

Mekern, V.N., Sjoerds, Z., & Hommel, B. (2019). How metacontrol biases and adaptivity impact performance in cognitive search tasks. Cognition, 182, 251-259.

Hommel, B., & Colzato, L.S. (2017). The social transmission of metacontrol policies: Mechanisms underlying the interpersonal transfer of persistence and flexibility. Neuroscience and Biobehavioral Reviews, 81, 43-58.

3. The first attempt to relate exploitation and exploration to creativity tasks was also ignored:

Hills, T. T., Todd, P. M., & Goldstone, R. L. (2008). Search in external and internal spaces. Psychological Science, 19(8), 802–808.

4. The parameter is used to indicate exploitation and exploration are likely to be affected, if not confounded with risk taking. Given that risk-taking has an impact on creativity, this is a serious problem: Shen, W., Hommel, B., Yuan, Y., Chang, L. & Zhang, W. (2018). Risk-taking and creativity: Convergent, but not divergent thinking is better in low-risk takers. Creativity Research Journal, 30, 224-231.

5. It remains unclear what the ratio between optimal and non-optimal choices is supposed to represent theoretically/mechanistically. Given that this is the only parameter that shows relatively systematic effects, more theoretical foundation is necessary.

6. Divergent thinking was assessed by means of a non-established, self-developed task that doesn't seem to be validated. Why was AUT not used instead?

7. It remained unclear how the findings shown in tables 2 and 3 were generated. It seems that all available predictors were entered into a linear regression analysis. This should be clarified. If my guess is correct, we are facing the following problems: a) the predictors are obviously highly interrelated, at least in several cases, which is an absolute no go for regression analyses; b) this analysis can only assess linear effects, even though nonlinear functions seem to play a much more decisive role: e.g., Akbari Chermahini, S., & Hommel, B. (2012). More creative through positive mood? Not everyone! Frontiers in Human Neuroscience, 6:319. Akbari Chermahini, S., & Hommel, B. (2010). The (b)link between creativity and dopamine: Spontaneous eye blink rates predict and dissociate divergent and convergent thinking. Cognition, 115, 458-465.

8. It remains unclear which outcome suggests that "divergent thinkers pursue both exploitation and exploration simultaneously". The fact that the ratio is involved does not speak to this issue at all, as the values are likely to represent different trials.

9. The two models do not replicate each other. Given that they are theoretical status is unclear, this means that be don't know what the outcome means.

10. In the discussion, there are repeatedly Chinese characters mentioned. I had no idea where they come from. In the description of the RAT, Chinese characters were not mentioned, and I found it difficult to imagine an RAT in which Chinese characters play a role.

signed, Bernhard Hommel

Reviewer #3: Review of Effect of exploitation vs. exploration on Creativity in Q-learning by Tsutomu Harada.

In this study, the author investigates the relationship between exploration and exploitation as defined in the reinforcement learning paradigm and convergent and divergent thinking as measured with different creativity tests. To this end, the author first analyzed the behaviors of 113 subjects performing the Iowa Gambling Task, by modelling the learning process with two different versions of the Q-Learning algorithm. From this analysis, measures of exploration and exploitation are derived. Secondly, participants performed different creativity tasks, some assessing more convergent thinking and other divergent thinking. The hypothesis tested in the study is that convergent thinking is linked to exploitation whereas divergent thinking is linked to exploration, but counterintuitively, the author found opposite results. Indeed, divergent thinking was more associated with exploration and divergent thinking with exploitation.

The question investigated in this study (i.e. investigating computationally the cognitive mechanisms underlying creativity) is interesting and a wide range of data has been collected to answer this question. However, essential results and critical quality checks are missing in the current version of the paper, particularly in the IGT / Q-Learning part on which most of my major concerns focus.

Major Concerns

#1 The exploration and exploitation measures are directly linked to the subjects' performance on the task which compounds one condition (repeated choices between the 4 same decks of card) across 100 trials. I would be interested in seeing this performance. Do the subjects learn to pick the best card(s) eventually and if so when (at which trial) are they starting to select the best option(s) consistently? This aspect is critical, particularly as the task is quite long, if subjects learn which is the best option after 20 trials, it is very likely that the choices made in the 80 remaining trials will be defined as exploitation, and that would just be the results of subjects learning to play the task and not a strategy that could be opposed to exploration. Depending on this result, the author could consider doing a similar analysis focusing on the learning phase (before reaching the plateau) that is maybe occurring during the first 30/50 trials? This may not be necessary but showing the performance of subjects is the only way to make sure of that.

#2 In the study, two different versions of Q-learning are used to explain subjects' behaviors in the IGT but we are not given any indications about the quality of their respective fit. Do these models explain well the data and is one of these better than the other? This is a crucial point as the exploration/exploitation measures used in the regression analyses are directly based on the models' predictions. In addition, if one of the model explains parsimoniously better the data than the other, I doubt that a regression analysis based on the "worst" model would be meaningful enough. Can the author give a reason for doing such an analysis?

#3 Following my previous concern, I would like to know the author's justifications about the candidate models selection. The two models used are rather different, the most sophisticated one including two new essential features (i.e. an asymmetry in learning rates as well as dynamic learning rates). We can imagine a wider model space which would also include models with only one of these features and compare their predictions. In any cases, it would be interesting to get further details about the selection of these particular features and models.

#4 Regarding the model fitting, emerging methods are used to ensure that the parameters optimization and models comparison procedures are reliable in a given task (See Palminteri et al., The Importance of Falsification in Computational Cognitive Modeling, 2017). The latter consist in testing the fitting procedure on synthetic data obtained from the models of interest. A parameter recovery analysis allows to check if the fitting procedure is able to retrieve the generative parameters from simulated data whereas a model recovery analysis allows to check if the models comparison procedure is able to recognize the generative model of the data, in a given task. Such analyses would strengthen the modelling part of the paper, which is crucial as the exploration/exploitation measures used in the regression analyses depend on it.

#5 Exploration in the paper is rather extremely defined as the number of choices of the option with the lowest Q-value. That makes, for instance, a choice of the second worst option, not an exploratory one. Can the author justify this narrow definition of an explorative choice? And are the results different if a broader definition is used (e.g. any choice that is not exploitative is explorative. Or is explorative a choice of an option associated to one of the two lowest Q-values)?

#6 In the reinforcement learning literature, the temperature (or inverse temperature here) can be interpreted as a parameter governing the exploration/exploitation tradeoff. I am surprised not to see any attempt at correlating this parameter with either measures of exploration / exploitation or directly with scores of convergent and divergent thinking. Has the author explored this aspect?

Minor Concerns

#1 Page 6 of the manuscript, it is written that the average loss concerning the two disadvantageous decks is -$50, is it not -$25 instead?

#2 Concerning the second model, the dynamic learning rates seem not to be bounded between 0 and 1, is it the case? If it is, I think it would be important to consider adding details about parameters bounds.

#3 Page 10, last sentence of the Asymmetric time-varying Q-Model paragraph, equations references seem to be wrong, they should be 4 and 5 instead of 6 and 5.

#4 Pages 18 and 22, the numbers of stars in the legend of the table are erroneous. One star for p<0.1 and for p<0.05.

6. PLOS authors have the option to publish the peer review history of their article (what does this mean?). If published, this will include your full peer review and any attached files.

Reviewer #1: No

Reviewer #2: Yes: Bernhard Hommel

Reviewer #3: No

---

## [Author Response · Author response to Decision Letter 0]

7 May 2020

We greatly appreciate very useful comments and suggestions by reviewers. We believe the current revised manuscript responded to most of the reviewers' comments and requests. For details, please see the attached "Response to Reviewers".

---

## [Decision Letter · Decision Letter 1]

22 Jun 2020

The Effects of Risk-taking, Exploitation, and Exploration on Creativity

PONE-D-20-02635R1

Dear Dr. Harada,

We’re pleased to inform you that your manuscript has been judged scientifically suitable for publication and will be formally accepted for publication once it meets all outstanding technical requirements.

Kind regards,

Baogui Xin, Ph.D.

Academic Editor

PLOS ONE

Additional Editor Comments (optional):

Reviewers' comments:

Reviewer's Responses to Questions

**Comments to the Author**

1. If the authors have adequately addressed your comments raised in a previous round of review and you feel that this manuscript is now acceptable for publication, you may indicate that here to bypass the “Comments to the Author” section, enter your conflict of interest statement in the “Confidential to Editor” section, and submit your "Accept" recommendation.

Reviewer #1: All comments have been addressed

2. Is the manuscript technically sound, and do the data support the conclusions?

Reviewer #1: Partly

3. Has the statistical analysis been performed appropriately and rigorously? 

Reviewer #1: N/A

4. Have the authors made all data underlying the findings in their manuscript fully available?

Reviewer #1: No

5. Is the manuscript presented in an intelligible fashion and written in standard English?

Reviewer #1: Yes

6. Review Comments to the Author

Reviewer #1: Based on the reviewers comments and suggestions the authors have considerably revised the manuscript, changed analyses, hypothesis and interpretation of their findings.

In the modified manuscript, the authors align their reanalysis on the hypothesis and suggestions made by the Reviewer #2 suggesting that it is risk-seeking attitudes but not exploration-exploitation trade-off that positively influences diverging thinking in creativity. In light of this new hypothesis the authors have also revised their computational model by integrating utility function into the Q-learning model to characterize risk-seeking or risk-averse distortions of outcomes.

I believe that authors have thoroughly reconsidered their manuscript although I have concerns with obtained findings and would be cautious with the interpretations. Firstly, obtained results stands clearly as a post-hoc interpretation of the data and their exploratory nature should be acknowledge and discussed. Secondly, most of the obtained effects are relatively weak (with p-value < 0.1 or 0.05) raising questions about their replicability.

Furthermore, sometimes authors do interpret the non-significant results (page 20 of the revised manuscript: “In the pooled sample, the risk aversion index μ exerted a negative effect whereas the risk seeking index ν had a positive effect on divergent thinking. This implies that participants with learning convergence behaved in a risk-seeking manner, regardless of the gains or losses”. In the Table 2 the effect of parameter μ is negative but does not reach significance so this statement is misleading.

Overall, I still think that the paper is weakly convincing but considering the overall interest of the subject it might be published as an exploratory paper in the field bridging instrumental learning and creativity testing.

Minor comments:

- The authors completely eliminated any model comparison and do not justify with their data whether adding utility function in the Q-learning model (which comes at a cost of 2 additional free parameters) actually accounted for the data better than the simple Q-learning model without risk-taking component.

- Text on page 22 of the revised manuscript does not match data in the Table 3: “The dependent variable was the convergent thinking score. As in Table 2, columns (1) and (2) in the table show the results for the pooled sample, columns (3) and (4) refer to the successful subsample, and columns (5) and (6) show the results for the unsuccessful subsample”.

7. PLOS authors have the option to publish the peer review history of their article (what does this mean?). If published, this will include your full peer review and any attached files.

Reviewer #1: No

---

## [Editor Report · Acceptance letter]

15 Jul 2020

PONE-D-20-02635R1 

The Effects of Risk-taking, Exploitation, and Exploration on Creativity 

Dear Dr. Harada:

I'm pleased to inform you that your manuscript has been deemed suitable for publication in PLOS ONE. Congratulations! Your manuscript is now with our production department. 

Kind regards, 

on behalf of

Professor Baogui Xin 

Academic Editor

PLOS ONE